# *"They are not HIV treatments drugs; they are preventive drugs (PrEP)"*. Experiences of PrEP uptake among vulnerable adolescent girls and young women in Tanzania

**Magreth Thadei Mwakilasa**[1,2]*, **Alexander Mwijage**[1], **Stella Mushy**[3], **Maryam Amour**[4], **Nathanael Sirili**[5], **Evaline Maziku**[6], **Samwel Likindikoki**[7], **Emmanuel Balandya**[8], **Gideon Kwesigabo**[9], **Benson Kidenya**[10], **Stephen E. Mshana**[11], **Eligius Lyamuya**[12], **Blandina Mmbaga**[13], **Bruno Sunguya**[4], **John Bartlett**[14]

1 Department of Medical Botany, Plant Breeding and Agronomy, Muhimbili University of Health and Allied Sciences, Dar es Salaam, Tanzania, 2 School of Nursing, Midwifery and Health System, University College of Dublin, Dublin, Ireland, 3 Department of Community Health Nursing, School of Nursing, Muhimbili University of Health and Allied Sciences, Dar es Salaam, Tanzania, 4 Department of Community Health, School of Public Health and Social Sciences, Muhimbili University of Health and Allied Sciences, Dar es Salaam, Tanzania, 5 Department of Development Studies, School of Public Health and Social Sciences, Muhimbili University of Health and Allied Sciences, Dar es Salaam, Tanzania, 6 HIV Prevention Unit, Ministry of Health, Dodoma, Tanzania, 7 Department of Psychiatry, School of Clinical Medicine, Muhimbili University of Health and Allied Sciences, Dar es Salaam, Tanzania, 8 Department of Physiology, Muhimbili University of Health and Allied Sciences, Dar es Salaam, Tanzania, 9 Department of Epidemiology and Biostatistics, Muhimbili University of Health and Allied Sciences, Dar es Salaam, Tanzania, 10 Department of Microbiology, Catholic University of Health and Allied Sciences, Mwanza, Tanzania, 11 Department of Biochemistry and Molecular Biology, Catholic University of Health and Allied Sciences, Mwanza, Tanzania, 12 Department of Microbiology and Immunology, Muhimbili University of Health and Allied Sciences, Dar es Salaam, Tanzania, 13 Faculty of Medicine, Kilimanjaro Christian Medical University College, Moshi, Tanzania, 14 Duke Global Health Institute, Duke University, Durham, NC, United States of America

* magreth.mwakilasa@ucdconnect.ie

**Data Availability Statement:** All relevant data are within the paper and its Supporting Information files.

## Abstract

### Introduction

HIV poses a significant global health concern, affecting adolescents among other populations. This is attributed to various vulnerabilities including biological factors, gender inequalities and limited access to comprehensive sexual and reproductive health services in sub-Saharan Africa. In Tanzania, adolescent girls, and young women (AGYW) face double the risk of HIV infection compared to their male counterparts. The introduction of pre-exposure prophylaxis (PrEP) in early 2018 brought hope for changing the HIV cascade in the country. However, numerous challenges still hinder PrEP uptake. Therefore, this study explored experiences of PrEP uptake among vulnerable AGYW in Tanzania.

### Methods

This study employed a phenomenological qualitative approach; 52 semi-structured interviews were carried out between May to November 2022 in the selected healthcare facilities in Tanzania. The study adopted inductive-deductive thematic analysis guided by the Social Ecological Model (SEM) to elicit the views of AGYW aged 15–24. Nvivo software was utilised to organise data.

**Funding:** Research activities reported in this publication was supported by the Fogarty International Centre of the National Institutes of Health under Award Number 1R25TW011227-01. The content is solely the responsibility of the authors and does not necessarily present the official views of the National Institutes of Health. The funders had no role in study design, data collection and analysis, decision to publish, or preparation of the manuscript.

**Competing interests:** I have read the journal's policy and the one author of this manuscript have the following competing interests. Prof. Bruno Sunguya, a co-author of this publication, held the position of Director of Research and Publication and a chair of the Institutional Review Board (IRB) at Muhimbili University of Health and Allied Sciences. Although Prof. Bruno Sunguya did not take part in the review or approval process for the study mentioned in this manuscript and abstained from any discussions or decisions regarding it, his role as the Director of Research and Publication at the university required him to sign the ultimate ethical approval document. The disclosure of this potential conflict of interest is done to guarantee transparency in the study process. All other remaining authors declare no conflicts of interest. This does not alter our adherence to PLOS ONE policies on sharing data and materials.

**Abbreviations:** AGYW, Adolescent Girls and Young Women-AGYW; PrEP, Pre- Exposure Prophylaxis; WHO, World Health Organization; NASHCoP, National AIDS, STIs, and Hepatitis Control Programme; KVP, Key and Vulnerable Populations; HIV, Human Immunodeficiency Virus; MoH, Ministry of Health; STIs, Sexual Transmitted Infections; SEM, Social Ecological Model; SRH, Sexual reproductive health; HCF, Health Care Facility; HCP, Health Care Provider.

## Results

This study has uplifted barriers and facilitators on PrEP uptake among AGYW in Tanzania. The barriers are categorized at individual, interpersonal, and institutional levels. The individual level barriers included pre-requisites for initiating PrEP, disbelief in the effectiveness of PrEP, interference of refill hours with working hours, financial constraints, and adherence to the pills. The interpersonal level barriers included misconceptions about PrEP pills, and labelling of PrEP users. The institutional level barriers included inadequate privacy, PrEP drug stockout, being turned away by health care facilities (HCF), long waiting times, and distance to the HCF. Facilitators included factors at individual level (experienced benefit of PrEP, adequate PrEP knowledge, having multiple partners, perceived risk due to the nature of the work, PrEP ensuring privacy), interpersonal level (support from social networks), and institutional level (Free availability of PrEP, receiving refill reminders).

## Conclusions

To overcome barriers to PrEP uptake among AGYW, it is crucial to develop multi-level interventions that consider personal, social, and structural factors hindering PrEP uptake. Implementing strategies like prioritizing off-site PrEP delivery and expanding community outreach for PrEP awareness can help dispel misconceptions and enhance uptake.

## Introduction

Improving HIV prevention, especially among at-risk populations like Adolescent Girls and Young Women (AGYW), is a critical priority in global public health. Pre-exposure prophylaxis (PrEP) is a promising HIV prevention tool that offers hope to reduce new infections [1]. PrEP empowers AGYW to have control of their own health without seeking negotiation from their sexual partner to use condoms [2]. Despite significant achievements in HIV prevention and treatment, UNAIDS reported that in 2023, there were 4,000 new HIV infections each week among AGYW aged 15–24 globally, of these, 3,100 were from Sub-Saharan Africa (SSA) [3,4]. Moreover, there is a threefold higher vulnerability to HIV infection among AGYW compared to their male peers in SSA [3]. This trend is evident in Tanzania, where incidences of HIV infection are higher among AGYW aged 15–24 than men of the same age group [5].

In Tanzania, the first PrEP demonstration projects were carried from May 2018 to March 2019, in 34 councils in Dar es Salaam, Mwanza, Kagera, Shinyanga, Songwe, Ruvuma, Iringa, Mbeya and Njombe [6]. Following the successful project completion, the government and stakeholders started small scale national wide rollout of PrEP in healthcare facilities located in areas mapped by Ministry of Health, National AIDS, STIs, and Hepatitis Control Programme (NASHCoP) that have larger concentrations of at risk populations [6]. The geographical selection in the rollout was targeted because while all AGYW are at risk of HIV infection, the HIV epidemic impact varies based on geographical location and socioeconomic conditions [7]. Currently PrEP is available in all 26 regions of the country in selected healthcare facilities (HCF), and there are more that 396,787 PrEP initiators [8]. In Tanzania, PrEP is available for Female Sex Workers (FSW), Men who have sex with Men (MSM), People who inject drugs (PWID), HIV negative partners of sero-discordant couples, and vulnerable AGYW [9,10]. The Tanzania PrEP implementation framework define vulnerable AGYW as those who meet specific vulnerability criteria. These criteria include not using condoms consistently with their

most recent partner in the past three months, having a current STI or an STI infection within the last three months, and receiving cash or items in exchange for sex in the last six months. Additionally, AGYW who are victims of sexual violence with an ongoing risk of repeated incidents or who are in sexual relationships with individuals who inject drugs (PWID) or HIV-positive partners are also considered vulnerable [6].

Despite efforts to boost PrEP uptake, hurdles persist for AGYW. Previous SSA studies conducted among adolescent and young people in SSA identified factors such as increased HIV risk perception, desire for HIV-negative status, negative condom attitudes, peer influence, social support, education, awareness of PrEP effectiveness and safety, and convenient service access free of charge [2,11–20]. Even among individuals with a strong interest in PrEP, key barriers to uptake persist including limited knowledge [20–23], distance to healthcare facilities [21], daily pill burden, low HIV risk perception, timing complexities, concerns about side effects and safety, fear of HIV testing, lack of social support, confidentiality issues, clinic waiting times, health worker attitudes, medication stockouts, the stigma associated with HIV-positive status and perceived promiscuity [2,18,19,21,24–27]. Importantly, access to PrEP among AGYW is significantly influenced by country-specific healthcare systems, policies and legal restriction for access health care services without parents/guardians in diverse SSA contexts [28]. In Tanzania, the legal age for accessing healthcare with parental guidance is typically 18; however, this can vary based on individual circumstances. Minors under 17 who are emancipated living independently, with friends, or with partner/husband may access services without parental approval or consent.

While the promise of PrEP in curbing the HIV epidemic is evident, research on its uptake among AGYW in Tanzania after the nationwide rollout remains limited. Most prior research in the country focused in demonstration projects [15,18,29]. In the demonstration project, PrEP services are more structured with ongoing service assessments, a condition not replicated in the nationwide PrEP rollout. Therefore, the PrEP uptake experienced in the demonstration project may differ in the real world. Prior studies often focused on limited at-risk populations, and generalised findings from all key and vulnerable populations may overlook distinctive challenges faced by AGYW. Therefore, this study explored barriers and facilitators for PrEP uptake among vulnerable AGYW in selected health care facilities after Tanzania's nationwide PrEP rollout. This study's findings can inform the development of more effective interventions to increase PrEP uptake among AGYW in Tanzania and beyond.

## Methods

### Study design

This study employed an explorative phenomenological qualitative design to gain a deeper understanding of the perspectives of AGYW on their experiences with PrEP use. Phenomenology is particularly suited to exploring individual lived experiences and the meanings individuals attach to those experiences [30]. This approach was ideal for this study because it aimed to capture the rich and varied perspectives of AGYW regarding PrEP uptake, including their motivations, challenges, and overall experiences within the healthcare system. The research was conducted in health care facilities (HCFs) located in Dar es Salaam (Ubungo and Ilala districts) and Tanga (Korogwe Municipality) in Tanzania. These specific regions, districts, and HCFs were chosen purposefully. Socioeconomic deprivation in neighbourhoods with high concentrations of vulnerable AGYW, the number of individuals currently using PrEP, existing HIV prevalence patterns, and the presence of HCFs offering PrEP services all informed this purposeful selection.

The study participants were AGYW aged 15 to 24 years who had experience with PrEP, including both current and past users. This includes individuals who are currently using PrEP, have ever used PrEP in the past, or have stopped using PrEP. To be eligible for the study, participants had to be receiving PrEP services at one of the participating HCFs. This focus on AGYW already engaged with the healthcare system for PrEP allowed for the exploration of factors influencing their experiences. According to the Tanzania framework for PrEP implementation, AGYW where considered to be at risk populations and eligible for PrEP if they do not use condoms consistently with partners, have sexually transmitted infections (STIs) or in the last three months received cash for sex, are victims of sexual violence, or in a sexual relationship with persons who inject drugs (PWID) or HIV negative partner [6].

## Sampling strategy

The study employed a purposive sampling strategy to choose participants, in which individuals were recruited with the aid of health care providers (HCPs) and peer navigators in specific study areas. HCP and peer navigators informed people about the study, and recruiting ads were placed in care and treatment centres (CTCs) throughout the study locations. Those who voluntarily agreed or were interested to participate, the researcher perused to read research information sheet to participants. Participants between the ages of 18 and 24 were required to give their informed consent, and those who were aged 15 to 17 who were emancipated or considered mature minors signed an informed consent [31].

## Data collection

In depth Interviews (IDIs) was used to collect data, IDIs were conducted in locations where participants felt comfortable, primarily in their homes, workplaces, friends' dwellings, or peer navigators' spaces. In some cases, interviews were conducted at HCFs based on participant preference. The IDIs were carried out in Kiswahili, the official language of Tanzania, by interviewers and participants fluent in that language. The duration of the interviews ranged from 30 minutes to one hour. Informed consent to record the IDIs was obtained verbally and in writing. To ensure a safe and supportive environment for participants to share their experiences, interviews were conducted privately with no other individuals present. Participant recruitment and data collection occurred between 30-05-2022, and 04-11-2022. During this period, a total of 52 in-depth, face-to-face interviews were conducted with AGYW.

The research team consisted of four members: two women and two men. The principal investigator, MTM (Assistant Research Fellow, MSc), and three research assistants, MSM (Social Scientist, MSc), GN (Research Assistant, BA), and OP (Project Assistant, MSc), all possess significant expertise in qualitative research. We acknowledge that our own experiences, gender identities, cultural backgrounds, and biases can influence researcher's interactions with participants and the data analysis process. Researcher biases, including selection and social desirability bias, can distort interactions with participants and skew data analysis by favouring certain outcomes and shaping interpretations based on preconceived beliefs. To mitigate this, we underwent a two-day online training session focused on maintaining researcher reflexivity throughout the study. This training involved group discussions and activities to critically examine our own backgrounds, values, and assumptions that might influence data collection and interpretation.

The interview schedule had sequenced semi-structured questions categorised into four distinct components. The initial set of questions pertained to introductory matters, while the subsequent section delved into topics concerning the acceptance of PrEP and the underlying motivations for its usage, followed by the participants' experiences in obtaining PrEP from

HCFs, and lastly, a set of concluding questions. Subsequently, the research team convened regular visual meetings to analyse the emerging patterns and determine the point at which no new information was being generated in each entry HCF. At this juncture, the team proposed shifting their focus to another HCF within the catchment area. Once it was confirmed that no additional data was being collected during the IDIs, the data collection process was concluded. Furthermore, no personal details about the participants were gathered to guarantee their anonymity. To safeguard the confidentiality of the participants, all gathered data was securely saved on a computer that requires a password for access. This computer is exclusively available to the researcher and the supervision team.

## Data analysis

Thematic analysis was used to analyse data, following six stages outlined by Braun and Clarke [32]. The Nvivo Software version 1.7.1 was utilized for the organization of data such as coding, categories creation and themes arrangement. The audio recorded IDIs were transcribed verbatim in Swahili and then translated into English and coded by the researcher and 3 research assistant who collected the data. Before coding, the three research assistants ensured the accuracy of the translations by cross-referencing English version with the original Swahili transcripts. Transcripts were anonymized, and the use of pseudonyms was adopted in renaming the transcripts, to ensure that there is no personal identifying information. Inductive line-by-line coding was conducted in Nvivo and after completion and going through the codes with AM and SM it became apparent that going forward the Social Ecological Model (SEM) of health will be appropriate to present our results [33], we therefore adopted deductive methods going forward. A codebook was developed by MTM and with the guidance of AM, SM and NS the generated codes were discussed with team members and then grouped into broader categories based on the observations that reflect vital issues identified by the interviewee. The two themes of barriers and facilitators were organized based on level in the SEM of health that are individuals, interpersonal and institutional.

## Ethical statement

The Muhimbili University of Health and Allied Sciences Institutional Review Board granted ethical approval for this study, Ref. MUHAS-REC-05-2022-1150. Permission to access the study sites was obtained from the President's Office and the Regional Administration and Local Government Tanzania (PO-RALG), the Regional Medical Officer (RMO), the District Medical Officer (DMO), and hospital management. Informed consent to participate in the study was obtained verbally and in writing.

## Results

A total of 52 AGYW participated in the study. The majority, 37 participants (71.2%), were aged between 20 and 24 years. Most participants, 36 (69.2%), resided in Dar es Salaam. At the time of the study, none of the participants had enrolled in school, but 31 (59.6%) reported having started or completed secondary education. Additionally, 29 (55.8%) reported using both condoms and PrEP for HIV prevention 29(55.8%). Most identified as single and self-employed. The majority accessed PrEP services at healthcare centres, with 37 (71.2%) reporting this source. **Table 1** below provides further demographic details.

Participants in this study identified several barriers limiting PrEP uptake and facilitators that can help improve the use of PrEP services among peers in HCF. The identified barriers and facilitators are reported based on factors in the SEM of health, which include personal factors like the knowledge, attitudes, behaviour, and self-concepts of AGYW about PrEP that

**Table 1. Demographic characteristics of vulnerable adolescent girls and young women PrEP.**

| Variable name | Categories | N (%) |
|---|---|---|
| **Age** | 15–19 | 15(28.8) |
| | 20–24 | 37(71.2) |
| **Region of Residence** | Dar es salaam | 36(69.2) |
| | Tanga | 16(30.8) |
| **District of residence** | Ilala | 17(32.7) |
| | Korogwe | 16(30.8) |
| | Ubungo | 19(36.5) |
| **Education Level** | Started or completed primary | 21(40.4) |
| | Started or completed secondary | 31(59.6) |
| **Marital Status** | Single | 47 (90.4) |
| | In relationship/married/cohabiting | 5 (9.6) |
| **Occupation** | Employed | 13(25.0) |
| | Self employed | 33(63.5) |
| | Unemployed | 6(11.5) |
| **HIV prevention method** | Both condom and PrEP | 29(55.8) |
| | PrEP only | 23(44.2) |
| **Duration of using PrEP** | Less than 6 months | 37(71.2) |
| | 7 months and more | 15(28.9) |
| **Facility Type** | Health centre | 37(71.1) |
| | Hospital | 15(28.9 |

affect their uptake. Interpersonal factors include the influence of formal and informal social networks and support systems such as work, family, and friends on AGYW PrEP uptake. Institutional factors are the factors that impact AGYW PrEP users while they are at the facility where they collect the pills. Fig 1 below display a summary of themes, sub-themes, and codes arranged following the SEM of health.

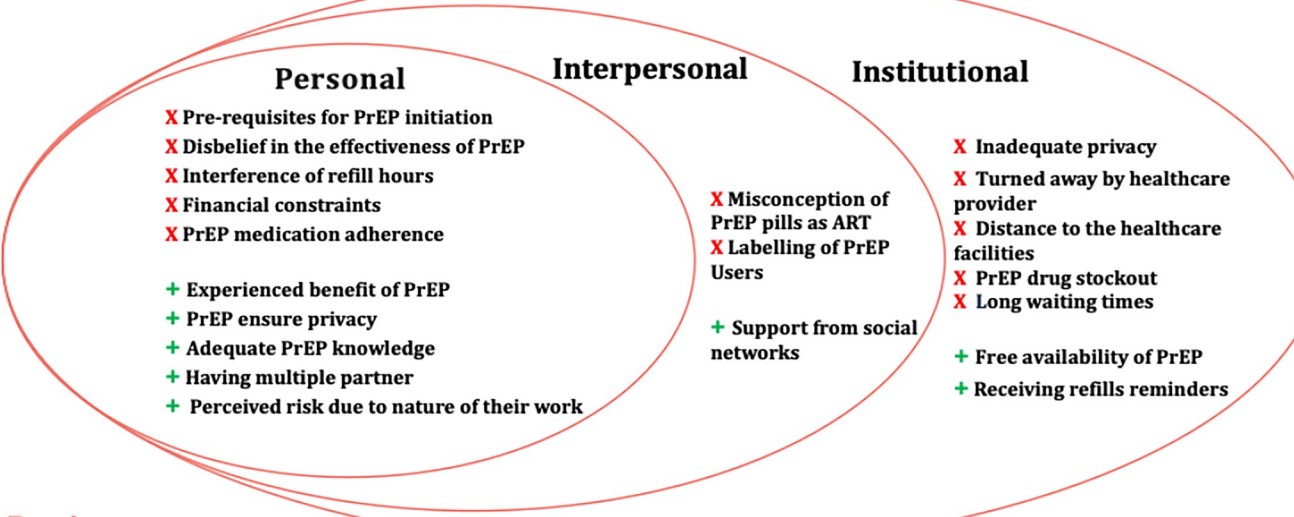

**Fig 1. A Social Ecological Model (SEM) of barriers and facilitators for PrEP uptake among adolescent girls and young women in Tanzania.**

## Barriers for PrEP uptake in HCF among AGYW

**Personal factors.** The study findings indicate that individual factors were identified as the main barriers hindering uptake to PrEP among AGYW who are at risk for HIV. These reported self-perceived barriers encompass pre-requisites for PrEP initiations, disbelief in the effectiveness of PrEP, interference of refill hours with working hours, financial constraints, and PrEP medication adherence, as discussed in detail below.

**The prerequisites for initiating PrEP.** The participants noted that the mandatory requirement of HIV testing prior to PrEP initiation is a barrier for several AGYWs intending to initiate PrEP use. The primary factor contributing to the participants' reticence that was revealed during the interviews was their apprehension about knowing their HIV status. One participant explained her hesitation experiences of PrEP initiation.

"*When they told me that I need to be tested first, I wanted to stop reaching decisions of using PrEP because I thought what would happen if I was tested and then I was found to have HIV infection. That is it, testing was giving me a lot of fear*" (P1, Age 17, Dar es Salaam)

**Disbelief effectiveness of PrEP.** Participants reported that peers are sceptical of the efficacy of PrEP in providing protection against HIV, hence they do not access the services as explained below;

*They don't believe it can prevent HIV, but PrEP is used in small amounts; a few percent is not much. I think because they don't know, they don't believe that the preventive medicine (PrEP) can work and that it prevents HIV; that is, they do not believe it works.* (P38, Age 24, Tanga)

Another participant who already uses PrEP, further elaborates on how she does not believe PrEP provides protection, and therefore they continue to use PrEP and condoms together as explained further by one participant;

"*Despite that, I had education about PrEP, but I could not stop using condoms because the first time they tell you about PrEP, it is as if they (HCF) are liars, and you think they are not useful. Now, after you go to the test, you are told good answers, and as you continue, you start to believe little by little that faith is coming*" (P20, Age 24, Dar es Salaam)

**Interference of refill hours with working hours.** Some participants reported that have not disclosed PrEP use to many in their immediate support system such as family members, friends, colleagues and employers. Due to the latter many find it difficult when asking permission to visit the HCF, since they cannot say they are going for PrEP some participants have reported being denied permission to visit the health centre;

"*because for a while I was stuck at work, because you know at work at other times, the boss does not let you leave just like that*" (P45, Age 23, Tanga)

Other participants further reported that due to the scheduling conflicts between PrEP refill hours and work hours, there were some time conflicts which made them unable to visit the HCF for PrEP refills;

"*It has happened before because here, where I work, there are times when it becomes difficult to go out, so I cannot go to get PrEP services, although there are times when I have permission from work and the drugs are not available*" (P9, Age 17, Dar es Salaam)

**Financial constraints.**   Several participants indicated that in order to reach the HCF, they were required to utilise various available modes of transportation, such as buses, motorbikes, or tuktuk (bajaji). However, they have acknowledged that they do face difficulties in obtaining the necessary travel fare and seeking PrEP services at the HCF. As a result of these circumstances, participants have reported several instances when they have chosen to defer or reschedule their trips to HCF for the purpose of accessing PrEP treatments, as documented below;

*I was unable to go for refills because I did not have transportation fare, and the health facility is far from here* (P19, Age 20, Dar es salaam)

**Adherence to the pills.**   AGYW noted that adherence to take PrEP medication everyday was one of the concerns they had to consider before initiating PrEP. The burden of taking pills everyday prevented many from accessing PrEP. The perceived fear of long-term side effects of taking the pills everyday also was reported as a significant issue that limited many participants to initiate PrEP use;

"*For the first time it is not easy to agree with that thing (PrEP), it is not easy to directly accept PrEP to use it. . . .at the very beginning I was wondering, just asking myself the purpose of the medicine, how do you take the medicine every day*" (P22, Age 20, Dar es Salaam)

**Interpersonal barriers.**   Participants identified several barriers with their immediate social groups, support system such as workmates, family members and friends that discouraged them at first to access care in the HCF. Among the reported barriers at this level are misconceptions of PrEP pills as ART and the labelling of PrEP users.

**Misconceptions about PrEP pills.**   Some participants using PrEP expressed that several of their peers, friends, workmates, and significant others believe that PrEP is an ART that is used in the treatment of HIV rather than its intended purpose of prevention. The identified misconception also made some individuals worried and scared to initiate or continue with PrEP uptake at the beginning, because people in their social environment associated PrEP use with HIV.

"*I don't have friends who use it, my friends themselves think that the drugs are for HIV, so I lose them because they talk. . .they think ooh I don't know this and that, or you will find yourself arguing with them "ooh I'm healthy"*" (P43, Age 23, Tanga)

Moreover, participant revealed the misconceptions that peers have about PrEP, is due to lack of knowledge and appropriate information about PrEP in their communities;

"*It depends on the understanding of the person who will be understood and whether they agree with it or not, because some say that it is not something to prevent, but it is a person who has already experienced the effects and is using it (PrEP). Therefore, others are growing to know that these are drugs used by affected people and not preventive medicine. Many are still worried about preventive medicine*" (P42, Age 21, Tanga)

**Labelling of PrEP users.**   Participants identified that in their communities, when people learned that they use PrEP they label them as HIV positive. Moreover, they spread information in the community and people start pointing fingers at you as and tell you that you are HIV positive, as explained below by one participant.

"*There was a time when my friends came to my house and took a picture of my PrEP can and started announcing that I am affected because I use those pills (PrEP)*" (P45, Age 23, Tanga)

The labelling does not end with people learning that they are using PrEP. Additionally, several participants identified that in most of HCF they seek care both PrEP and ART (for HIV treatment) are offered in the same building, the CTCs. As they are seen by people going into the CTCs or meet someone, they know inside they are also being perceived and labelled as HIV positive. The latter has resulted in a reluctance to uptake PrEP services due to fear of being identified or labelled as HIV positive. One participant provides a detailed account of her personal experience in the next below;

"*These PrEP services are available at the same time and area, with those of people who use (HIV)anti-infection drugs, that's where the challenge arises, when you meet someone, from the same street, so they automatically assume that I am also taking HIV drugs of infection. That's where the trouble begins, they start spreading the information that I've met him somewhere he's taking HIV anti-retroviral drugs*" (P22, Age 20, Dar es Salaam)

Moreover, a few participants revealed that they are afraid to be seen going to HCF to take PrEP as use is associated with sexual promiscuity and HIV positivity;

"*I am just afraid to say that maybe people will think bad of me, if it's not bad I'll tell them too! They will see me like me, maybe this girl is selling herself, that's why she's using PrEP. I'm not free to because when I come to (HCF) I'm tight and I have to hide myself because if people see me, they will also know that I have HIV virus (AIDS)*". (P52, Age 19, Tanga)

**Institutional barriers.** Participants in this study have reported several barriers that they face at the institutional level when accessing PrEP services in the HCF. These include inadequate privacy, PrEP drug stockout, long waiting times and distance to the HCF. These barriers are further elaborated below;

**Inadequate privacy.** Inadequate privacy is reported based on the area of HCF where PrEP services are offered, the location of the facility, and while waiting for services. Participants reported that PrEP services in the HCF are offered in the CTCs, an area that is commonly known for providing HIV treatment care for people who are HIV positive. Many identified that they feel uncomfortable accessing PrEP in the CTC, and in some situations, they send someone else to pick up their pills for the CTCs;

"*Another challenge that I see is that if someone like this comes to take medicine, someone knows that maybe you have come to take AIDS medicine…….. for example, when you enter here, people know very well that this person is going to take HIV treatment medicine (ARTs)… eenh of the unit, when it is not ARTs there are other preventative medicines (PrEP)*" (P44, Age 24, Tanga)

They further revealed that the CTC building within the HCF compound is located in an open location where they can easily be spotted by anyone who knows them. Apart from the latter, participants outlined that their privacy is still compromised even after entering the CTC building, arguing that PrEP users and HIV-positive individuals all receive care in the same area or room. Their major concern is the fear of being identified by someone they know;

"*The environment is bad because when we take PrEP in the same area where people with HIV are treated; that's what they do." So if someone passes by from a distance, they will say you are in the same group, so I don't like it. I'm not free. The interaction is good, but I don't like it. Staying with people with HIV, even though we're all humans. It might even happen that your person (referring to your man) saw you sitting there (CTCs); he will not care; he will perceive that you are already infected. Yes, they stopped giving you money for food. He ran away from you while you were there to take preventive medicine that protected him too.*" (P43, Age 23, Tanga)

**PrEP drug stockout.** Participants highlighted there is infrequent availability of PrEP in HCF and this discourages many from continuing to seek PrEP services. While other participants further describe that they visited the HCF up to twice-thrice attempting to either initiate PrEP or refill pills on all occasions they were told PrEP is not available;

"*But we went there (HCF) three times, and each time we went, the medicines were finished. First, when we went, there were so many of us, so we found a long queue. When it was our turn, they told us the medicine was finished and we should return the following week. When that date arrived, we went and were told again to return on a certain date, but we did not go back*" (P9, Age 17, Dar es salaam)

They reported that HCP informed them that the inconsistences in the availability of PrEP is due to the expired medications, and they were waiting for new drug stock and generally PrEP is scarce in the country, and they should patiently wait for them to be available.

**Turned away by HCF.** Participants outlined that there are some few occasions that they run out of PrEP prior to their refill date, and they approached the HCP seeking pills but were denied without any regard for the time and transportation expenses they had incurred. The denial was primarily due to the fact that it was not a designated day for PrEP distribution or because they did not have a scheduled appointment. One participant recounts her experiences below;

"*The day I went was not the day of those services. I was told to come on a certain day, and I actually went and was given care.I felt bad because I looked at the medicines, I had run out. . . .I had only one pill left. Since I knew the importance medicines are important, I felt bad for missing taking them(PrEP)*" (P40, Age 22, Tanga)

**Long waiting times.** The amount of time spent waiting for PrEP services at the HCF was a major concern for AGYW we interviewed. They described that the long waiting times were primarily due to a mismatch between the number of HCPs in the CTCs and the number of clients or patients they needed to serve. AGYW recounted experiences of long queues and instances where only one nurse was available to serve the CTCs. They also noted that the waiting times were particularly long during PrEP initiation, as this process requires screening and numerous tests before administering PrEP.

"*I see that the services are not good, it is challenging, because there are delays in getting the service. The lack of nurses causes us to take a long time, but if there were more nurses, the procedures would be done more quickly*" (P3, Age 17, Dar es salaam)

AGYW further uncovered that the waiting is exceptionally long when going for PrEP initiation, as they are required to be screened and run numerous tests before administering PrEP.

**Distance to the HCF.**   Several participants indicated that the proximity to the healthcare facility (HCF) did not pose a significant issue, either it was short, or they had access to several transportation choices, particularly when they had bus fares. There are instances in which some AGYW found to be without the necessary funds to pay for bus transportation. Consequently, they resorted to walking for a duration of between 45 minutes to 2 hours, depending on the distance, in order to reach the HCF. As a result, there were multiple occurrences in which the individuals had a delay in obtaining a refill of PrEP medication;

"*There are those who get services in nearby places and there are those who come from distant places. There are those who spend even two hours on the journey just to reach the health center*" *(*P5, Age 19, Dar es Salaam).

Additionally, AGYW delineated concern of accessing PrEP in HCF that are located within a shorter distance to their vicinity, with them being scared of being labelled and being identified as using PrEP;

"*When you look at the environment, for example, I can't be living here and go to the nearest healthcare facility to take PrEP, people will misunderstand me, you know?. . . . . . . .but going to healthcare facility far away it's becoming normal here. Around hear (Place of residence) people will start to judge that I have started to eats HIV treatment drug (ART) and is not preventive medicine (PrEP)*" (P20, Age 24, Dar es Salaam)

## Facilitators for accessing PrEP among AGYW in HCF

**Personal facilitators.**   Among personal facilitators for accessing PrEP services in HCF that were reported by participants in this study included experienced benefit of PrEP, accessibility of PrEP, adequate PrEP knowledge, having multiple partners, perceived risk due to the nature of the work and PrEP ensure privacy.

**Experienced benefit of PrEP.**   Several participants mentioned the experienced protective benefit of PrEP against HIV is what motivates them to continue utilizing PrEP;

"*I just liked to use it because I love it and care a lot about my health. That is, I don't want to get HIV infection. They (HCP) told me to use those drugs because they are good. They told me that according to the environment I live in, those drugs are good because I can have sex with a person with the HIV virus and I will be safe*" (P8, Age 16, Dar es Salaam)

**Adequate PrEP knowledge.**   Several participants said that their utilization and access to PrEP were improved as a result of receiving adequate information about PrEP from HCP and friends;

"*Mmmh. . . actually what attracted me was that I was satisfied with the information the doctor gave me while giving me medicine (PrEP), so I saw that it was the right way for me*" (P14, Age 19, Dar es Salaam)

**Having multiple partners.**   Some of participants conveyed that their decision to adopt PrEP was primarily influenced by their engagement in sexual intercourse with many non-permanent partners. The participants expressed that the usage of PrEP provided them the ability

to engage in sexual activity with several partners without being aware of or concerned about their HIV status, particularly in situations where their partner declined to use a condom. Many participants in the study acknowledged that their primary motivation for engaging in sexual relationships with multiple partners was the financial support they received. This financial incentive played a key role in shaping their relationship choices. One of the participants provides additional clarification in the subsequent section;

"*"I'm motivated to continue using PrEP because I'm myself, that is, I don't have one partner, Eeenh, that is, I don't really have a permanent"* (P37, Age 24, Tanga)

A few participants also pointed out that their access to PrEP services was facilitated by the fact that they have a relationship with people who are living with HIV, as explained by one participant;

"*What convinced me was because I was living with a partner who was living with HIV infection, so I was convinced because there was an element that specifically says that the person who is supposed to use PREP is me who lives with a partner who is infected"* (P51, Age 24, Tanga)

**Perceived risk due to the nature of the work.** Certain participants employed in high-risk environments, such as bars, male saloon, and those engaged in sex work, have acknowledged that their perception of the hazardous nature of their work environment plays a role in shaping their inclination to continue accessing PrEP. The participants in the study indicated that they encounter situations at their workplace where they are enticed by financial incentives from clients. Consequently, they engage in sexual activities with these clients without being aware of their health status or utilizing protective measures such as condoms. Therefore, the utilization of PrEP serves to guarantee consistent protection in such instances;

"*What led me to say let me use it is because of the work we do we encounter many challenges when we are at work, we meet people we do not know, and we cannot test them to know if they are sick or healthy. So, when I got the information about these medicines (PrEP), I was grateful because they protect me, then before using I checked my health and found that I was fine so I used it with confidence, and I know that I am using it to protect myself"* (P9, Age 17, Dar es Salaam)

**PrEP ensured privacy.** Several participants in the study expressed that the absence of a requirement to disclose the use of PrEP, like the use of condoms, served as a facilitator for their continued access to PrEP. The participants expressed difficulties in engaging in discussions regarding condom usage and HIV testing with the numerous short-term partners they encountered. Therefore, many perceive the use of PrEP to ensure ongoing safeguard against HIV in these particular situations;

"I *can say these medicines (PrEP) are good because they give you the assurance of not getting an infection. The most interesting thing is that you can use these drugs even if your partner doesn't want to, because you can use them without him knowing, this is completely different from other methods like using condoms because you both have to agree"* (P6, Age 17, Dar es Salaam)

### Interpersonal facilitators

**Support from social networks.**   Several participants expressed it is the encouragement and influences that they get from their family members, friends and workmates that encourages them to continue accessing PrEP. Participants mentioned incidences where they received advice from friends to initiate use, to take the pills and reminder to go for PrEP refills from friends. Arguing the acquired support facilitated them to access PrEP services easily;

"*I heard people were talking on WhatsApp groups only, they were to be talking and talking. So, to me I found someone, and I went and asked him to explain to me about this because she is an adult and I told him then it's fine and I started using PrEP*" (P32, Age 17, Dar es Salaam)

"*but when I involved my friend, she encouraged me, I came back the next day and tested, and it was negative. That is when I found the strength to use PrEP*" (**P1, 17 years-Dar es Salaam**)

**Institutional facilitators.**   Participants indicated that institutional facilitators encouraging their continued uptake of PrEP include the availability of the medication at no cost and the provision of refills.

**Free availability of PrEP.**   Some participants mentioned that the fact that PrEP pills and related services are available for free was viewed as an opportunity and motivated many to continue accessing care/services. As reported by one participant below;

"*First, I saw that we are given for free, as well as that we are kept in a good environment of not being infected with the HIV virus whenever we have sex with an infected person*" (P3, Age 17, Dar es Salaam)

Other participants revealed to them what facilitated to continues access PrEP was close proximity to the HCF;

*I think it is easy and convenient to use PrEP probably because I am close to where I go to take my medicine. (*P46, Age 20, Tanga)

**Receiving refills reminders.**   According to participants, receiving refill reminders from HCP or their peer educators helped them to remember accessing PrEP services in the HCF. They outline the phone calls and text reminders from HCP close to the date of refills or on the day of refills played a crucial role in enhancing their utilization of PrEP services;

"*Something that makes it easier is sometimes they remind us of when the date is approaching, they call us. They send us a message as soon as the dates are approaching*" (P15, Age 21, Dar es Salaam)

"*There were still about three days left when I got a call, and he told me that the medicine was about to run out. If you're going to be a while, come and pick it up, don't wait until that date arrives, come pick it up, we'll be there for you*" (P46, Age 20, Tanga)

Moreover, other participants mentioned that having the clinic card that contains information about their next visits to the clinic also was served as a useful reminder to go to the HCF for refills.

## Discussion

To our knowledge this is the first study to explore experienced PrEP uptake barriers and facilitators among AGYW in Dar es Salaam and Tanga, Tanzania after the completion of PrEP rollout in the country. Consistent with previous research conducted prior to PrEP the roll-out in SSA that utilized SEM [15,18,19], this study highlighted that there are still multiple factors at the personal, interpersonal and institutional levels that continue to affect the uptake of PrEP among AGYW. Although previous studies conducted before and during the demonstration project were helpful in steering the rollout of PrEP in SSA and specifically in Tanzania, a more comprehensive understanding of the barriers and facilitators that aid the process is crucial for increasing the uptake of PrEP among AGYW in the general population. The rationale behind this is that the uptake of PrEP not only provides protection against HIV for individuals but also enhances their involvement with health services, including SRH, mental health support, and counselling [6].

The findings of this study are similar to investigations carried out in Kenya, Uganda, Malawi, and South Africa that reported individual hurdles to PrEP uptake, including testing before starting PrEP, questions about its efficacy, refill hours conflicting with work hours, and medication adherence issues [2,11,16,18,34]. Differently from prior studies, this study found that financial constraints may act as a barrier to PrEP uptake among study participants. Participants reported missing HCF appointments due to transportation issues, leading them to postpone PrEP refill appointments. This finding underlines the need for more HCF and community delivery to reach different demographics. Furthermore, additional factors that contribute to uptake at this level encompass participants' heightened perception of HIV danger [14,35,36], unlike in previous studies this study found a high risk associated with having multiple sexual relationships and occupational characteristics, such as employment in a bar or male saloon. Additional participants in the study note that they were inspired to take PrEP as it preserves their privacy in a way that they do not have to negotiate or disclose use with their partners and are in control of whether to use it or not. These findings should be considered when designing interventions to promote positive thoughts and PrEP use at the individual level to overcome barriers.

Misunderstanding PrEP as ART is a barrier to PrEP uptake on an interpersonal level. According to participants, peers, friends, co-workers, and the community viewed PrEP as ART, attributing the latter misunderstanding to public PrEP ignorance. This study's results are similar to an AGYW study in Uganda [16] and another involving adolescents and young people of both sexes in Uganda, Zimbabwe, and South Africa [18]. The following research found that participants confused PrEP with post-exposure prophylaxis (PEP), and that PrEP uptake is linked to a variety of terms and conditions [18]. Differences in misunderstandings regarding PrEP can be ascribed to disparities in the temporal and contextual circumstances under which the studies were conducted. Both studies were conducted within the framework of the structural project, with one study taking place before the roll-out of PrEP, and our study took place after the roll-out of PrEP in the general population. Additionally, the current study found that community and workplace colleagues characterise PrEP users as HIV positive or sexually promiscuous, which hinders uptake. This finding matches previous research [16,37,38]. The current study findings relate to earlier research because many African communities stigmatise HIV-related healthcare services due to cultural norms and a lack of factual information. This study, like others, found that social support from peers, friends, family, and colleagues helps people take PrEP [16,18]. The study also recognises the ongoing community's PrEP awareness campaigns targeting KVP, such as AGYW, FSW, MSM, and PWID. We emphasise the need to invest more in community-wide outreach activities to promote positive attitudes towards

PrEP among the general public. The latter is crucial to allow AGYW and other PrEP users to access services without fear of stigma or labelling.

At the institutional level, similar to past research, the study has identified several problems that hinder the uptake of PrEP. The current study reported instances of PrEP medicine stock-outs, situations where healthcare providers send patients away, and extended waiting periods in health facilities; these findings are consistent with previous research conducted in Uganda, Zimbabwe, South Africa, and Tanzania [16,18,19]. This relevance is particularly notable in resource-limited settings, where difficulties in accessing essential resources such as medications and healthcare staff in health facilities are common obstacles to seeking medical care. Nevertheless, in contrast to previous research, this study revealed the absence of physical privacy in the CTCs, causing AGYW to feel discouraged and uneasy about seeking care. This is because they fear being stigmatized as HIV positive, as CTCs are consistently linked with HIV-related stigma. This study emphasises the difficulty of incorporating PrEP into the current framework of HIV services, which is one of the approaches used in the implementation of PrEP. The second discovery indicates a significant necessity to contemplate the decentralization of PrEP services from the conventional CTCs in the country and prioritise the facility-led community delivery of PrEP.

Furthermore, like studies carried out in Uganda, one done in a rural area and another in an urban area emphasized that the proximity to the HCF was a hindrance to PrEP uptake [16,21]. The present investigation, carried out in the urban areas of Dar es Salaam and Tanga, likewise documented comparable results. In contrast to the previous study, this study examined distance from two different viewpoints. Some participants emphasized that the HCF is situated at a considerable distance, requiring them to travel or walk a significant distance to reach. Other participants expressed discomfort with accessing PrEP services at nearby HCF due to concerns about being labelled as HIV positive. The aforementioned findings suggest that there is a poor understanding of PrEP among the general population. Previous research has also highlighted the necessity of community awareness interventions to address the misunderstandings held by the public regarding PrEP [24,39,40]. The data also indicate that it is essential to augment the quantity of HCFs providing PrEP throughout the nation to address the issue of extended distances.

The provision of free PrEP was found to be an important factor in facilitating uptake of PrEP at the institutional level. This finding aligns with earlier research conducted among AGYW in SSA [18]. In contrast to other research, participants in this study contend that receiving refill reminders via mobile phone, either through a text message or a phone call from HCF, served as a significant incentive to continue their PrEP uptake. The latter emphasises the significance of promoting telemedicine and technology to retain younger individuals in PrEP care. Studies on healthcare and medication adherence, including among PrEP users, have found that the utilisation of mobile apps or technology significantly enhances patient retention in care [16,41,42]. These findings should be considered when exploring strategies to enhance PrEP usage and adherence among existing PrEP users.

A notable deficit of this study is its lack of generalizability to the entire population of AGYW residing in Tanzania. Given the vast size of the country, contextual factors in different regions can significantly impact AGYW's behaviour towards PrEP uptake. However, a notable advantage of this study is that it provides a perspective on the difficulties that have arisen as a result of the nationwide rollout of PrEP. The findings can provide crucial insights into the events occurring on the ground and promote policy focus on community-driven initiatives that focus on the stigmatization of PrEP. Furthermore, this study is limited by the presence of social desirability bias. Due to the lack of a precise metric for measuring the distance between the HCF and participants' houses, we had to depend on the participants' self-reported

information and their own interpretation of what constitutes a short or long distance. Nevertheless, we have provided training to researchers to equip them with the ability to ask additional inquiries that can help us comprehend the distance. These queries pertain to the time required to reach the HCF by foot, by car, and using other modes of transportation. Additionally, the interview guide was not piloted beforehand. However, following the initial day of data collection, the research team convened to deliberate on the difficulties encountered with the instruments and determine whether any modifications were necessary. Nevertheless, the instruments proved to be efficient, so no alterations were implemented.

## Conclusion

This study has uncovered numerous barriers and facilitators at different levels of the SEM, providing insight into the intricate factors that influence uptake of PrEP among AGYW. The study emphasizes the need to derive multi-level interventions that are more accommodating for AGYW and advocating for comprehensive approaches such as prioritising out-of-facility PrEP delivery and increasing community outreach to raise awareness of PrEP, which in turn will reduce PrEP misconceptions and contribute to the increase in PrEP uptake.

## Supporting information

**S1 File. Consolidated criteria for reporting qualitative research checklist.**
(DOCX)

**S2 File. Demographic profile data.**
(XLSX)

**S3 File. Selected quotes from qualitative transcripts.**
(DOCX)

**S4 File. Interview guide.**
(DOCX)

## Acknowledgments

The authors wish to thank the seniors and junior mentors within the Community of Young Research Peers (CYRPs) consortium for their endless guidance and expertise in conducting this study. A special thanks to my research assistants Mohamed Seif Mohamed (MSM), Getrude Matee (GM), and Optatus Kasogela (OK) for their dedicated efforts and attention to details throughout the study. I would like to extend my deepest appreciation to my research mentees, Silvano Bairon and Lilian Katabalo, for their enthusiasm in learning how to carry out qualitative research and their contribution to this research. I am confident that their passion for research will lead to continued success.

## Author Contributions

**Conceptualization:** Magreth Thadei Mwakilasa, Alexander Mwijage, Maryam Amour, Nathanael Sirili, Evaline Maziku, Samwel Likindikoki, Bruno Sunguya.

**Data curation:** Magreth Thadei Mwakilasa.

**Formal analysis:** Magreth Thadei Mwakilasa, Alexander Mwijage, Stella Mushy, Nathanael Sirili.

**Funding acquisition:** Magreth Thadei Mwakilasa, Emmanuel Balandya, Gideon Kwesigabo, Benson Kidenya, Stephen E. Mshana, Eligius Lyamuya, Blandina Mmbaga, Bruno Sunguya, John Bartlett.

**Investigation:** Magreth Thadei Mwakilasa.

**Methodology:** Magreth Thadei Mwakilasa.

**Project administration:** Magreth Thadei Mwakilasa, Maryam Amour.

**Resources:** Magreth Thadei Mwakilasa.

**Software:** Magreth Thadei Mwakilasa.

**Supervision:** Alexander Mwijage, Maryam Amour, Nathanael Sirili, Evaline Maziku, Samwel Likindikoki, Emmanuel Balandya, Bruno Sunguya.

**Validation:** Magreth Thadei Mwakilasa, Maryam Amour, Nathanael Sirili, Gideon Kwesigabo, Benson Kidenya, Stephen E. Mshana, Blandina Mmbaga, Bruno Sunguya, John Bartlett.

**Visualization:** Eligius Lyamuya.

**Writing – original draft:** Magreth Thadei Mwakilasa.

**Writing – review & editing:** Magreth Thadei Mwakilasa, Alexander Mwijage, Stella Mushy, Maryam Amour, Nathanael Sirili, Emmanuel Balandya, Gideon Kwesigabo, Eligius Lyamuya, Bruno Sunguya, John Bartlett.

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
