## [Decision Letter · Decision Letter 0]

29 Sep 2024

PONE-D-24-16479“They are not HIV treatments drugs; they are preventive drugs (PrEP) ”. Experiences of using PrEP among vulnerable adolescent girls and young women in TanzaniaPLOS ONE

Dear Dr. Mwakilasa,

Thank you for submitting your manuscript to PLOS ONE. After careful consideration, we feel that it has merit but does not fully meet PLOS ONE’s publication criteria as it currently stands. Therefore, we invite you to submit a revised version of the manuscript that addresses the points raised during the review process.

The manuscript is well conducted and written. I would like to request the authors to address the very minor comments by Reviewer 3.

We look forward to receiving your revised manuscript.

Kind regards,

Swarnali Goswami

Academic Editor

PLOS ONE

“Research activities reported in this publication was supported by the Fogarty

International Centre of the National Institutes of Health under Award Number

1R25TW011227-01. The content is solely the responsibility of the authors and does

not necessarily present the official views of the National Institutes of Health.”

“I have read the journal's policy and the one author of this manuscript have the following competing interests. Prof. Bruno Sunguya, a co-author of this publication, held the position of Director of Research and Publication and a chair of the Institutional Review Board (IRB) at Muhimbili University of Health and Allied Sciences. Although Prof. Bruno Sunguya did not take part in the review or approval process for the study mentioned in this manuscript and abstained from any discussions or decisions regarding it, his role as the Director of Research and Publication at the university required him to sign the ultimate ethical approval document. The disclosure of this potential conflict of interest is done to guarantee transparency in the study process. All other remaining authors declare no conflicts of interest.”

5. In this instance it seems there may be acceptable restrictions in place that prevent the public sharing of your minimal data. However, in line with our goal of ensuring long-term data availability to all interested researchers, PLOS’ Data Policy states that authors cannot be the sole named individuals responsible for ensuring data access (http://journals.plos.org/plosone/s/data-availability#loc-acceptable-data-sharing-methods).

Reviewers' comments:

Reviewer's Responses to Questions

**Comments to the Author**

1. Is the manuscript technically sound, and do the data support the conclusions?

Reviewer #1: Yes

Reviewer #2: Yes

Reviewer #3: Yes

2. Has the statistical analysis been performed appropriately and rigorously? 

Reviewer #1: Yes

Reviewer #2: N/A

Reviewer #3: N/A

3. Have the authors made all data underlying the findings in their manuscript fully available?

Reviewer #1: Yes

Reviewer #2: No

Reviewer #3: Yes

4. Is the manuscript presented in an intelligible fashion and written in standard English?

Reviewer #1: Yes

Reviewer #2: Yes

Reviewer #3: Yes

5. Review Comments to the Author

Reviewer #1: The manuscript was developed in a professional manner and suits the criteria for publication by the journal. The statistical analysis is high sounding where there is use of qualitative software. There is however need to revise minor grammatical errors.

On the methods section, the authors just started by mentioning this is a qualitative study withing giving the study design and that should be stated as well.

Reviewer #2: For the Results section, I recommend to include a table with Participants’ characteristics? It can include Age, Sex, School level, PrEP status, etc.

The rest of my comments I have submited in a word document.

Reviewer #3: Thank you for the opportunity to review this manuscript. The authors present a well conducted and written qualitative study investigating barriers and facilitators to the use of PrEP amongst young women in Tanzania. I would like to commend the authors on their systematic approach to the study and the study write-up. I have a few very minor comments for consideration:

1. In the introduction can the authors expand on what is meant by "vulnerable" AGYW? Given the listing of previous groups it seems redundant to say vulnerable here?

2. In the study design section can the authors clarify if the participants had to be current users of PrEP, or if they could be past users, provided that they had used the HCP previously.

3. The data collection section mentions that the interviews were conducted in Kiswahili. Can the authors please provide some information on how the quotations (or perhaps interviews) were translated into english? I think this can be added to the methods section very easily.

4. Line 414-416 - this seems to be repeated at the end of the paragraph just before the quotation.

5. Line 432-434 - I am not sure that I understand the last part of the quotation?

6. PLOS authors have the option to publish the peer review history of their article (what does this mean?). If published, this will include your full peer review and any attached files.

Reviewer #1: **Yes: **Fungai Mudzengerere

Reviewer #2: **Yes: **Maria de los Angeles Cabrera Escobar

Reviewer #3: **Yes: **Meagen Rosenthal

---

## [Author Response · Author response to Decision Letter 0]

23 Oct 2024

Rebuttal letter responding to editor and reviewer comments.

Editor comments

1. Please ensure that your manuscript meets PLOS ONE's style requirements, including those for file naming. Please ensure that your manuscript meets PLOS ONE's style requirements, including those for file naming. The PLOS ONE style templates can be found at

Authour Feedback: In the title page the author has added details of the corresponding author and email address as provided in the guideline. The manuscript structure has followed the guideline provided in the respected provided link

If this statement is not correct you must amend it as needed

Authour Feedback: The amended role of funder statement has being edit to included the recommended sentence and the role of funder statement has being added to the cover letter for the journal team to change the online submission form on my behalf.

Authour Feedback: An updated competing interests statement has being added to the cover letter your cover letter for the journal to change the online submission form on my behalf

4. We note that you have indicated that there are restrictions to data sharing for this study. PLOS only allows data to be available upon request if there are legal or ethical restrictions on sharing data publicly. For more information on unacceptable data access restrictions, please see http://journals.plos.org/plosone/s/data-availability#loc-unacceptable-data-access-restrictions.Before we proceed with your manuscript, please address the following prompts:

4a) If there are ethical or legal restrictions on sharing a de-identified data set, please explain them in detail (e.g., data contain potentially identifying or sensitive patient information, data are owned by a third-party organization, etc.) and who has imposed them (e.g., a Research Ethics Committee or Institutional Review Board, etc.). Please also provide contact information for a data access committee, ethics committee, or other institutional body to which data requests may be sent.

4b) If there are no restrictions, please upload the minimal anonymized data set necessary to replicate your study findings to a stable, public repository and provide us with the relevant URLs, DOIs, or accession numbers. For a list of recommended repositories, please see https://journals.plos.org/plosone/s/recommended-repositories. 

You also have the option of uploading the data as Supporting Information files, but we would recommend depositing data directly to a data repository if possible.We will update your Data Availability statement on your behalf to reflect the information you provide

Authour Feedback: The minimal data required to reproduce the results of the manuscript has been uploaded along with the manuscript. As supporting file 2 and 3. 

5. In this instance it seems there may be acceptable restrictions in place that prevent the public sharing of your minimal data. However, in line with our goal of ensuring long-term data availability to all interested researchers, PLOS’ Data Policy states that authors cannot be the sole named individuals responsible for ensuring data access (http://journals.plos.org/plosone/s/data-availability#loc-acceptable-data-sharing-methods).Data requests to a non-author institutional point of contact, such as a data access or ethics committee, helps guarantee long term stability and availability of data. Providing interested researchers with a durable point of contact ensures data will be accessible even if an author changes email addresses, institutions, or becomes unavailable to answer requests.Before we proceed with your manuscript, please also provide non-author contact information (phone/email/hyperlink) for a data access committee, ethics committee, or other institutional body to which data requests may be sent. If no institutional body is available to respond to requests for your minimal data, please consider if there any institutional representatives who did not collaborate in the study, and are not listed as authors on the manuscript, who would be able to hold the data and respond to external requests for data access? If so, please provide their contact information (i.e., email address). Please also provide details on how you will ensure persistent or long-term data storage and availability.

Authour Feedback: The minimal data required to reproduce the results of the manuscript has been uploaded along with the manuscript. As supporting file 2 and 3

6. Please review your reference list to ensure that it is complete and correct. If you have cited papers that have been retracted, please include the rationale for doing so in the manuscript text, or remove these references and replace them with relevant current references. Any changes to the reference list should be mentioned in the rebuttal letter that accompanies your revised manuscript. If you need to cite a retracted article, indicate the article’s retracted status in the References list and also include a citation and full reference for the retraction notice

Authour Feedback: There was one duplicated in the reference list, and that was adjusted by removing the duplicate reference.

Reviewer 1 Comments

1. Reviewer #1: The manuscript was developed in a professional manner and suits the criteria for publication by the journal. The statistical analysis is high sounding where there is use of qualitative software. There is however need to revise minor grammatical errors.On the methods section, the authors just started by mentioning this is a qualitative study without giving the study design and that should be stated as well

Authour Feedback: The research design has been added in line see the clean manuscript line 137-138.

Review 2 Comments 

2. Reviewer #2: For the Results section, I recommend to include a table with participants’ characteristics? It can include Age, Sex, School level, PrEP status, etc. The rest of my comments I have submited in a word document

Authour Feedback: Also In the clean manuscript, I added figure, instead of a table that was in the initial submission to summarize the list of themes and sub-themes reported in the results section using SEM. I’m not sure if this is acceptable to the journal. If not, I can revert to the original version with a table instead of a figure

Reviwers 3 Comments

Reviewer #3: Thank you for the opportunity to review this manuscript. The authors present a well conducted and written qualitative study investigating barriers and facilitators to the use of PrEP amongst young women in Tanzania. I would like to commend the authors on their systematic approach to the study and the study write-up. I have a few very minor comments for consideration

1. In the introduction can the authors expand on what is meant by "vulnerable" AGYW? Given the listing of previous groups it seems redundant to say vulnerable here?

Authour Feedback: This information has been added see clean manuscript introduction section paragraph two line 98-105.

2. In the study design section can the authors clarify if the participants had to be current users of PrEP, or if they could be past users, provided that they had used the HCP previously

Authour Feedback: This information has been added see clean manuscript study design section paragraph two-line number 149-150

3. The data collection section mentions that the interviews were conducted in Kiswahili. Can the authors please provide some information on how the quotations (or perhaps interviews) were translated into english? I think this can be added to the methods section very easily.

Authour Feedback: This information has been added see clean manuscript see methods section, data analysis sub-section line 210-214

4. Line 414-416 - this seems to be repeated at the end of the paragraph just before the quotation.

Authour Feedback: The section has been re written and the repeat information has been removed see result section line 433-439

5. 5. Line 432-434 - I am not sure that I understand the last part of the quotation?

Authour Feedback: The quotation is made clear to communicated what participant inferred see result section line 440-442

6. PLOS authors have the option to publish the peer review history of their article (what does this mean?). If published, this will include your full peer review and any attached files

Authour Feedback: Yes 

Comments in the word document comments

1. 44 The introduction 

45 of pre-exposure prophylaxis (PrEP) brought hope for changing the HIV cascade in the 

Could you include the year PrEP was introduced in Tanzania. It gives an idea of the time that people have been exposed to PrEP.

Authour Feedback: This information has been added see abstract section, introduction sub-section line 48-50.

2. 94 identified factors such as increased HIV risk perception, desire for HIV-negative status, 

95 negative condom attitudes, peer influence, social support, education, awareness of PrEP 

96 effectiveness and safety, and convenient service access free of charge (2,11–20).

Authour Feedback: This information has been re-written to be more clear see clean manuscript see introduction section, paragraph 3 line 106-111

3. 97 barriers to PrEP uptake, even for highly interested individuals, include limited knowledge Suggestion: Even for individuals with a high HIV risk exposure

104 without parents/guardians in diverse SSA contexts (30). Can you conclude this sentence explaining how is in Tanzania.

Authour Feedback: This information has been added to be more clear see clean manuscript see introduction section, paragraph 3 line 118-122

4. 167 biases? Which type of biases?

Authour Feedback: This information about research biases has been elaborated to be more clear see clean, method section , data collection subsection, paragraph 2 line 184-189

5. 212 Most participants in this study were 20–24, resided in Dar es Salaam, and were out of 213 school with a secondary education level

Include numbers or % here.

Ex. Of the 52 participants interviewed, 44 were between 20 and 24 year sold

Authour Feedback: A table with demographic characteristics has been added see clean manuscript line 234-243

6. 304 limits them to accessing and initiating PrEP freely. The identified misconception also made 

305 some worried and scared to initiate PrEP use at the beginning because people in their 

306 social environment associated PrEP use with HIV;

This sentence is good but too long, can you use coma or re-phrase it

Authour Feedback: This sentence has been re written and split by a comma to make it short see clean manuscript line 335-337

7. 460 particularly in situations where their partner declined to use a condom. Furthermore, a 

461 significant number of participants in the study recognized that their motivation for 

462 engaging in sexual relationships with several partners was primarily driven by the financial 

463 support they received from these partners.

Authour Feedback: This sentence has been re written and to make it short see clean manuscript results section line 492-495

8. These two finding are similar but yield opposite results. Can you explain this. 294 Interpersonal barriers 

507 Interpersonal facilitators 

These two finding are similar but yield opposite results. Can you explain this.

Authour Feedback: We report on the interpersonal barriers that prevent AGYW from taking PrEP. However, they communicated that receiving social support from their social network would facilitate their uptake of PrEP. Thus, while the findings are at the same interpersonal level, barriers are the negative factors that hinder uptake, and facilitators are the positive factors that motivate uptake.

9. 519 The institutional facilitators reported by participants for continuing to access PrEP 

520 included the factors of PrEP accessibility and receiving refills reminders, as elaborated

Suggestion of re-phrace it: “Participants reported that institutional facilitators encourage them continued PrEP use by ensuring accessibility and providing refill reminders”. You can re-phrase it to make it clear.

Authour Feedback: The suggestion has been integrated in the see clean manuscript, result section line 554-555

10. 599 At the institutional level, similar to past research, the study has identified several problems 

600 that hinder the uptake of PrEP.

 just added a comma

Authour Feedback: The suggestion to better improve the sentence by adding a comma can be reflected in the clean manuscript discussion section line number 634.

11. 600 These include instances of PrEP medicine stock-outs, 

601 instances when HCP sends patients away, and extended waiting periods for care at HCF due 

602 to a disparity between the number of staff and PrEP users (17,19,20) .

This discussion is in which context, in Tanzania or local context or other countries?

Authour Feedback: The context of the discussion has been added to the clean manuscript discussion section line 635-638.

12. 602 The latter relevance 

603 can be attributed to the fact that in settings with limited resources, the problem of 

604 accessing resources like drugs and staff in HCF is a common obstacle to seeking medical 

605 attention

Authour Feedback: The context of the discussion has been added to the clean manuscript discussion section line 638-641

---

## [Editor Report · Decision Letter 1]

25 Oct 2024

“They are not HIV treatments drugs; they are preventive drugs (PrEP) ”. Experiences of PrEP uptake among vulnerable adolescent girls and young women in Tanzania

PONE-D-24-16479R1

Dear Dr. Mwakilasa,

We’re pleased to inform you that your manuscript has been judged scientifically suitable for publication and will be formally accepted for publication once it meets all outstanding technical requirements.

Kind regards,

Swarnali Goswami

Academic Editor

PLOS ONE
---

## [Editor Report · Acceptance letter]

29 Oct 2024

PONE-D-24-16479R1 

PLOS ONE

Dear Dr. Mwakilasa, 

I'm pleased to inform you that your manuscript has been deemed suitable for publication in PLOS ONE. Congratulations! Your manuscript is now being handed over to our production team.

Kind regards, 

on behalf of

Dr. Swarnali Goswami 

Academic Editor

PLOS ONE